# Provably Safe Online Multi-Agent Navigation
# in Unknown Environments

**Zhan Gao\*  Guang Yang\*  Jasmine Bayrooti  Amanda Prorok**

University of Cambridge

**Abstract:** Control Barrier Functions (CBFs) provide safety guarantees for multi-agent navigation. However, traditional approaches require full knowledge of the environment (e.g., obstacle positions and shapes) to formulate CBFs and hence, are not applicable in unknown environments. This paper overcomes this issue by proposing an Online Exploration-based Control Lyapunov Barrier Function (OE-CLBF) controller. It estimates the unknown environment by learning its corresponding CBF with a Support Vector Machine (SVM) in an online manner, using local neighborhood information, and leverages the latter to generate actions for safe navigation. To reduce the computation incurred by the online SVM training, we use an Imitation Learning (IL) framework to predict the importance of neighboring agents with Graph Attention Networks (GATs), and train the SVM only with information received from neighbors of high 'value'. The OE-CLBF allows for decentralized deployment, and importantly, provides provable safety guarantees that we derive in this paper. Experiments corroborate theoretical findings and demonstrate superior performance w.r.t. state-of-the-art baselines in a variety of unknown environments.

**Keywords:** Decentralized Multi-Agent Navigation, Unknown Environment, Support Vector Machine, Graph Attention Learning, Control Barrier Function

## 1  Introduction

Multi-agent systems have garnered significant attention for applications such as distributed aerial surveillance [1, 2], search and rescue [3, 4], and warehouse transportation [5, 6], where safe and efficient navigation is a key enabling technique. In this work, we focus on the problem of decentralized multi-agent navigation with dynamical constraints. Different from classic path finding problems [7, 8, 9], we consider both state and control spaces in the continuous domain, and aim to develop controllers that generate feedback control inputs based on local neighborhood information. This problem can be formulated as a sequence of real-time quadratic programming with control barrier functions (CBFs) and control Lyapunov functions (CLFs), where CBFs encode safety for collision avoidance and CLFs encode convergence for goal reaching. The combination of CBFs and CLFs has been widely used for safety-critical controls [10, 11, 12]. While providing safety guarantees, traditional CBF-based approaches require full knowledge of the environment, e.g., positions and shapes of obstacles, to formulate a closed-form expression for the safety function [13, 14]. Such information is not available when navigating agents in unknown environments, which happens commonly for decentralized multi-agent systems that have limited observability of the real world.

Our goal is to synthesize decentralized controllers that navigate agents in unknown environments with provable safety guarantees. This problem is challenging because it is difficult to formulate the CBF of an unknown environment. Using sensing input, such as vision and LiDAR, is a potential solution, which, however, remains an open question as traditional control-based methods lack the

---

*Equal contribution.

8th Conference on Robot Learning (CoRL 2024), Munich, Germany.

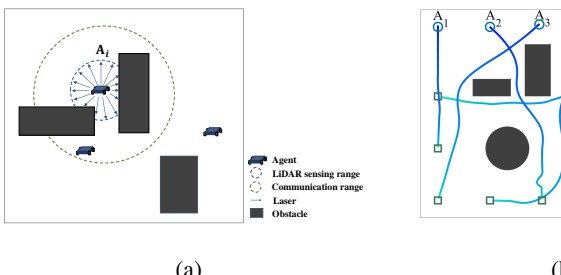

(a)    (b)

Figure 1. (a) Local agent $A_i$ senses surrounding environment using LiDAR scans (blue circle) with lasers (blue arrays), and communicates with neighbors within communication range (green circle). (b) An example of our method applied in an unknown environment with different obstacle positions, sizes and shapes. Blue circles are initial positions and green squares are destinations. Obstacles are black, and blue-to-green lines are agent trajectories. The color bar represents time scale, showing that no collision occurs.

expressiveness to deal with rich sensing models. For example, it is unclear how to handcraft differentiable CBFs with a discrete number of observations (e.g., visual images or LiDAR readings). Learning-based methods have been shown successful for sensing-feedback control tasks [15, 16], and recent works have used neural networks to approximate CBFs and CLFs to design observation-feedback controllers to navigate in unknown environments [17, 18, 19]. However, these approaches often sacrifice safety guarantees of the controller. Moreover, they follow an offline supervised learning framework, which suffers from distribution shifts when the deployment environments are vastly different from the training environments; hence, resulting in significant performance degradation.

In pursuit of the aforementioned goal, we propose a sensing-feedback controller for decentralized multi-agent navigation in unknown environments, by utilizing an *online learning-based approach* without requiring any prior knowledge about the environment. Our contributions are as follows:

**(i)** We learn the environment CBF with a support vector machine (SVM), using LiDAR readings received from neighboring agents, and incorporate the SVM-based CBF into a quadratic programming (QP) controller for safe navigation in unknown environments. Moreover, we train SVM in an online manner with instantaneous LiDAR readings, which adapts to any new (unseen) environments.

**(ii)** We leverage graph attention learning to quantify neighbors' importance and perform training only with observations received from important neighbors, to reduce online computation.

**(iii)** We provide safety guarantees for the proposed learning-based method by leveraging the accuracy bound of SVM and the state invariance of CBFs.

**(iv)** We evaluate our method in a variety of navigation scenarios and show a significant improvement over other leading approaches. We also incorporate an exploration mechanism into the practical implementation of our method, so that agents are encouraged out of local minima.

**Related Work.** CBF-based methods have been widely used for safety-critical controls [10, 11, 12]. Traditional approaches often assume a centralized setting with global information of the environment or agents to formulate safety functions [13, 14]. Distributed CBFs consider the partial observability of agents to provide decentralized safe navigation [20, 21], while these works address only inter-agent collision avoidance but not obstacles. The work in [22] leverages reinforcement learning to prevent deadlocks when deploying distributed CBFs. However, it still requires prior knowledge of obstacle configurations in the environment to handcraft CBFs and is not applicable in unknown environments. Moreover, the hand-designed analytical CBF form may not capture the environment accurately even with global information, leading to inevitable performance degradation.

Learning-based approaches provide a handle to use sensing input for controller synthesis, which is a potential solution for safe navigation in unknown environments. Most existing works conduct the safety check of the learned sensing-based controller *post hoc*, where the controller is synthesized and then independently verified. The work in [23] uses adversarial testing to assess safety and searches for the most likely input that causes the vision-based controller to fail, while [24] trains a generative adversarial network to map states to image observations and uses the latter to check the safety of the

concatenated controller network. Several works attempt to incorporate the safety consideration into the controller synthesis. The works in [25, 26, 27] learn a perception map from observations and use it to design controllers, but these approaches make strong assumptions. For example, they assume that the environment (including all obstacle locations) is known prior to deployment, limiting its generalization to unknown environments. The authors in [28, 29] leverage SVM and Gaussian process (GP), respectively, to approximate the CBF of the unknown environment with LiDAR readings. However, all these works consider a single-agent system, and [28] requires discretizing the state space to verify the safety. The work in [30] attempts to extend the GP-based CBF to multiple agents, while it ignores inter-agent communication with a lack of system-level consideration and its experiment only involves three agents. More closely related works [17, 18, 19] use neural networks to approximate CBFs and CLFs to design observation-feedback controllers. These neural network based methods enjoy scalability benefits and are versatile w.r.t. the input space. However, they follow an offline supervised learning framework, which leads to significant performance degradation if deploying environments are vastly different from training ones. Moreover, the learned barrier certificate is an approximation of the true one and it is difficult to provide theoretical safety guarantees.

## 2   Preliminaries

**System dynamics.** Consider a multi-agent system of $n$ agents $\mathcal{A} = \{A_i\}_{i=1}^n$. The dynamics of agent $A_i$ is of the form

$$\dot{\mathbf{x}}_i = f(\mathbf{x}_i) + g(\mathbf{x}_i)\mathbf{u}_i \tag{1}$$

with $\mathbf{x}_i$ the state, $\mathbf{u}_i$ the control input, $\dot{\mathbf{x}}_i$ the time derivative of $\mathbf{x}_i$, and $f(\mathbf{x}_i)$, $g(\mathbf{x}_i)$ the flow vectors.

**Control Lyapunov function (CLF).** The CLF is designed to encode the goal reaching constraint, i.e., the saistification of CLF constraints guarantees agents converging to their designated goals. In this work, we define an exponentially-stabilizing control Lypaunov function [31] as a positive definite function $V_i(\mathbf{x}_i) : \mathbb{R}^n \mapsto \mathbb{R}$ [31] such that there exists a constant $\epsilon \geq 0$ with

$$\inf_{\mathbf{u}_i \in \mathbb{U}_i} [\mathcal{L}_f V_i(\mathbf{x}_i) + \mathcal{L}_g V_i(\mathbf{x}_i) u_i + \epsilon V_i(\mathbf{x}_i)] \leq 0, \tag{2}$$

where $\mathcal{L}_f V_i(\mathbf{x}_i) := \frac{\partial V_i(\mathbf{x}_i)}{\partial \mathbf{x}_i} f(\mathbf{x}_i)$ is the Lie derivative of the continuously differentiable function $V_i(\mathbf{x}_i)$ along dynamics $f(\mathbf{x}_i)$, $\mathbb{U}_i$ the control space of agent $A_i$ and $\epsilon$ controls the convergence rate.

**Control barrier function (CBF).** The safety requirement of agent $A_i$ is equivalent to that a continuously differentiable function $h_i(\mathbf{x}_i) : \mathbb{R}^n \mapsto \mathbb{R}$ and its derivative w.r.t. time satisfy

$$h_i(\mathbf{x}_i) \geq 0, \ \dot{h}_i(\mathbf{x}_i) + \alpha(h_i(\mathbf{x}_i)) \geq 0, \ \text{with} \ \dot{h}_i(\mathbf{x}_i) = \mathcal{L}_f h_i(\mathbf{x}_i) + \mathcal{L}_g h_i(\mathbf{x}_i)\mathbf{u}_i, \tag{3}$$

where $\mathcal{L}_f h_i(\mathbf{x}_i), \mathcal{L}_g h_i(\mathbf{x}_i)$ are the Lie derivatives of $h_i(\mathbf{x}_i)$, and $\alpha(\cdot)$ is a class $\mathcal{K}$ function that determines how strictly we enforce safety. We define the higher-order CBF [12] with relative degree one in this paper, and the satisfaction of (3) ensures *forward invariance* of the state trajectory [12].

## 3   Problem Formulation

Consider a multi-agent system $\mathcal{A}$ in an unknown environment $\mathcal{E}$, i.e., the agents have no information about neither the obstacle positions nor the obstacle shapes. The agents are initialized at positions $\mathbf{S} = [\mathbf{s}_1, ..., \mathbf{s}_n]$ and tasked towards $\mathbf{G} = [\mathbf{g}_1, ..., \mathbf{g}_m]$. Each agent is equipped with a LiDAR scanner that senses its surrounding environment with a number of $L$ lasers and returns distances to the end-points of these lasers. The goal is to navigate the agents from $\mathbf{S}$ to $\mathbf{G}$ while avoiding collision with obstacles and other agents. Specifically, let $\mathbf{x}_i$ be the local state of agent $A_i$ that includes its position $\mathbf{p}_i$, velocity $\mathbf{v}_i$, destination $\mathbf{g}_i$ and LiDAR readings, and let $\mathbf{x}_i^{(0)}$ and $\mathbf{x}_i^d$ be the initial and target states. We consider a *decentralized* setting, where each agent senses its local state, communicates with its neighboring agents, and generates its own action with only neighborhood information – see Fig. 1a. Thus, the destination convergence is equivalent to the state convergence $\lim_{t \to T} \mathbf{x}_i^{(t)} = \mathbf{x}_i^d$ with $T$ the maximal time step. Let also $\mathcal{C}_{\text{safe}} \subseteq \mathcal{E}$ be the obstacle-free space that can be occupied by the agents. Collision avoidance is equivalent to the safety constraint $\mathbf{x}_i^{(t)} \in \mathcal{C}_{\text{safe}}$ for $t = 1, ..., T$. This allows to formulate the problem of safe decentralized multi-agent navigation.

**Problem 1** (Safe decentralized multi-agent navigation). *Given the multi-agent system $\mathcal{A}$ with dynamics (1), the initial and target states $\{\mathbf{x}_i^{(0)}\}_{i=1}^n$ and $\{\mathbf{x}_i^d\}_{i=1}^n$, find decentralized navigation policies $\{\pi_i\}_{i=1}^n$ that generate agent actions $\mathbf{U} = [\mathbf{u}_1^{(t)}, ..., \mathbf{u}_n^{(t)}]$ such that*

$$\lim_{t \to T} \|\mathbf{x}_i^{(t)} - \mathbf{x}_i^d\| = 0 \ \text{ and } \ \mathbf{x}_i^{(t)} \in \mathcal{C}_{\text{safe}}, \ \text{ for all } t = 1, \dots, T \text{ and } i = 1, \dots, n. \tag{4}$$

The first condition in (4) encodes the state convergence, i.e., navigation to the goals, and the second condition in (4) encodes state safety, i.e., collision avoidance. Problem 1 is challenging because (*i*) the safety set $\mathcal{C}_{\text{safe}}$ is not available as the environment is unknown; and (*ii*) the decentralized policies $\{\pi_i\}_{i=1}^n$ require generating agent actions only with local neighborhood information.

## 4 Methodology

To overcome these challenges, we propose to estimate the safety set $\mathcal{C}_{\text{safe}}$ by synthesizing the CBF of the unknown environment with LiDAR readings sensed by the local agent and received from its neighboring agents, and combine the latter with CLFs into a quadratic programming (QP) controller to solve Problem 1. Since it is difficult to formulate continuously differentiable CBFs with a discrete number of LiDAR readings, we leverage a learning-based approach to synthesize the environment CBF and perform training online to provide a real-time representation for the surrounding environment. To reduce the computation of online learning, we learn the importance of neighboring agents with graph attention networks (GATs), which have achieved success in a wide array of applications and allow for a decentralized implementation [32], and train with only LiDAR readings from important neighbors. More specifically, our method contains two main components.

**SVM-based CBF-CLF-QP navigation.** This component estimates the unknown environment by learning its corresponding CBF with online support vector machine (SVM) using instantaneous LiDAR readings at each time step. It then employs a CBF-CLF-QP controller to generate control inputs for the agents based on local neighborhood information, and integrates an exploration mechanism into the controller to avoid infeasibility of the QP controller during navigation.

**Attention-based computation reduction.** This component leverages the GAT to predict the importance of the neighboring agents' states in estimating the surrounding environment of the local agent, by following an imitation learning framework. With a cut-off threshold $\gamma$, it trains the SVM only with LiDAR readings of the neighboring agents whose attentions are larger than $\gamma$, to reduce the computation of online SVM training.

With these components, we propose an Online Exploration-based Control Lyapunov Barrier Function (OE-CLBF) controller that allows to conduct safe multi-agent navigation in unknown environments in a decentralized manner. It can be applied directly to any environment (i.e., any obstacle layout) without fine-tuning or pre-training, due to the nature of online learning – see Fig. 1b for an example. Moreover, it is capable of providing provable safety guarantees due to accuracy guarantees of the SVM and the state invariance of CBFs.

### 4.1 SVM-based CBF-CLF-QP Navigation

We leverage a CBF-CLF-QP controller to solve Problem (1) iteratively across time. Specifically, at each time step $t$, we re-formulate Problem 1 as a QP problem with CBF and CLF constraints

$$\begin{aligned}
\min_{\mathbf{u}_i^{(t)} \in \mathbb{U}_i, \delta_i^{(t)} \in \mathbb{R}} \quad & \|\mathbf{u}_i^{(t)}\|_2 + \xi(\delta_i^{(t)})^2 \\
\text{s.t. } & \mathcal{L}_f^{r_b} h_i(\mathbf{x}_i^{(t)}) + \mathcal{L}_g \mathcal{L}_f^{r_b-1} h_i(\mathbf{x}_i^{(t)}) \mathbf{u}_i^{(t)} + \alpha_{i,r_b}(h_i(\mathbf{x}_i^{(t)})) \geq 0, \\
& \mathcal{L}_f V_i(\mathbf{x}_i^{(t)}) + \mathcal{L}_g V_i(\mathbf{x}_i^{(t)}) \mathbf{u}_i^{(t)} + \epsilon V_i(\mathbf{x}_i^{(t)}) + \delta_i^{(t)} \leq 0, \text{ for all } i = 1, \dots, n
\end{aligned} \tag{5}$$

with $\delta_i \in \mathbb{R}$ a slack variable that determines how strictly we enforce the CLF constraint, $\xi \in \mathbb{R}^+$ a regularization parameter often set as one, and $\mathbb{U}_i$ the action space given physical constraints.

However, (5) cannot be formulated directly because the environment is unknown and thus, CBF constraints are not available. To overcome this issue, we learn the CBF of the unknown environment with LiDAR readings sensed by the agents. Specifically, the raw LiDAR reading is a distance $d_l \in [0, d_{\max}]$, with $d_{\max}$ the maximum scanning distance. When $d_l = d_{\max}$, there is no collision detected w.r.t. the $l$th laser and no obstacle along that direction within the sensing range – see Fig. 1a. We transform $d_l$ to the endpoint location of the laser $\mathbf{z}_l$ based on the agent position $\mathbf{p}$, and label $\mathbf{z}_l$ with a binary variable $y_l \in \{+1, -1\}$ where $+1$ indicates a detected obstacle and $-1$ otherwise.

At each time step $t$, agent $A_i$ obtains local LiDAR readings $\mathcal{Z}_i^{(t)} = \{(\mathbf{z}_{i,l}^{(t)}, y_{i,l}^{(t)})\}_{l=1}^L$ and communicates with its neighbors to collect nearby LiDAR readings $\{\mathcal{Z}_j^{(t)}\}_{j \in \mathcal{N}_i}$, where $\mathcal{N}_i$ is the neighbor set of $A_i$ within communication range. By considering the aggregated $\mathcal{Z}_i^{(t)}$ and $\{\mathcal{Z}_j^{(t)}\}_{j \in \mathcal{N}_i}$ as the training data, we can learn the safety function $h$ of the environment CBF in (5). Since LiDAR readings change across time steps as the agent moves, it requires online learning to capture a real-time up-to-date representation of the surrounding environment. As the label $y_{i,l}^{(t)}$ is binary, we propose to leverage SVM because of its fast training for online implementation and high accuracy for performance guarantees. Specifically, we consider the SVM classifier as $f_{\mathrm{SVM}}(\mathbf{z}_l) = \mathrm{sign}(\langle \boldsymbol{\omega}, \phi(\mathbf{z}_l) \rangle_{\mathcal{F}})$ with parameters $\boldsymbol{\omega}$ [33], where $\phi(\cdot)$ is a lifting function that maps the input data to some feature space $\mathcal{F}$ and $\langle \cdot \rangle_{\mathcal{F}}$ is some inner product defined in $\mathcal{F}$, and solve the optimization problem as [34]

$$\min_{\boldsymbol{\omega}} \quad \frac{1}{2}\|\boldsymbol{\omega}\|^2 \quad \text{subject to} \quad y_l \langle \boldsymbol{\omega}, \phi(\mathbf{z}_l) \rangle_{\mathcal{F}} \geq 1 \text{ for all } l. \tag{6}$$

By solving (6) in the dual domain to obtain the support vector set $\mathrm{N_{S.V.}}$ and its corresponding Lagrangian multipliers $\boldsymbol{\beta}$, we can estimate the safety function with the decision function of SVM as

$$h_{\mathrm{SVM}}(\mathbf{z}) = \sum_{l \in \mathrm{N_{S.V.}}} y_l \beta_l \langle \phi(\mathbf{z}_l), \phi(\mathbf{z}) \rangle_{\mathcal{F}}, \tag{7}$$

which measures how safe the position $\mathbf{x}$ is w.r.t. the obstacles. By substituting $h_{\mathrm{SVM}}$ into (5), we formulate an SVM-based CBF-CLF-QP controller that generates safe actions for each agent $A_i$ with online SVM-based CBFs learned from its local LiDAR readings $\mathcal{Z}_i^{(t)}$ and $\{\mathcal{Z}_j^{(t)}\}_{j \in \mathcal{N}_i}$. By selecting a differentiable lifting function $\phi(\cdot)$ for SVM, the safety function $h_{\mathrm{SVM}}(\cdot)$ and the resulting SVM-based CBF constraint are differentiable for the QP controller (5). Furthermore, an exploration mechanism is incorporated to resolve the potential infeasibility of the QP controller in challenging scenarios during navigation (e.g., non-convex safety regions) – see details in Appendix B.

## 4.2 Attention-based Computation Reduction

The SVM-based CBF-CLF-QP controller requires training an SVM online with instantaneous LiDAR readings at each time step $t$. However, it may be computationally expensive if the number of LiDAR readings, i.e., the number of neighboring agents, is large, since the time complexity of SVM training is quadratic w.r.t. the number of training samples. To reduce the online computation, we quantify the importance of neighboring agents by learning edge weights with a GAT and train the SVM only with LiDAR readings received from important neighbors. Specifically, consider the multi-agent system $\mathcal{A}$ as a graph $\mathcal{G}$, where the nodes are the agents $\{A_i\}_{i=1}^n$ and the edges are communication links. The graph structure is captured by an adjacent matrix $\mathbf{W}$, where $[\mathbf{W}]_{ij} \neq 0$ if there is a link between $A_i$ and $A_j$, and $[\mathbf{W}]_{ij} = 0$ otherwise. The value of $[\mathbf{W}]_{ij}$ represents how important the neighboring agent $A_j$ is w.r.t. the local agent $A_i$. The agent states are modeled by $\mathbf{X}$, which is a matrix whose $i$th row is the state of $A_i$. We define decentralized attention generators $g_i(\mathbf{x}_i, \{\mathbf{x}_j\}_{j \in \mathcal{N}_i})$ for $i = 1, ..., n$, which computes edge weights $\{[\mathbf{W}]_{ij}\}_{j \in \mathcal{N}_i}$ based on local neighborhood information. We then formulate the problem of graph attention learning as follows.

**Problem 2** (Graph attention learning). *Given the agent states $\mathbf{X}$, find decentralized generators $\{g_i\}_{i=1}^n$ that compute edge weights with local neighborhood information, to identify the importance of the neighboring agents to the local agent in synthesizing the CBFs of an unknown environment.*

Problem 2 is challenging because it is difficult to define the objective function for training, i.e., it is non-trivial to measure how well the generated attentions quantify the neighbors' importance, and

we propose to follow an imitation learning framework to solve this problem. In particular, let $\mathbf{x}_i = \{\mathbf{p}_i, \mathbf{v}_i, \mathbf{g}_i, \mathcal{Z}_i\}$ be the states of $A_i$. For the scenario without any concern of computation, each agent $A_i$ aggregates all available states $\mathbf{x}_i$ and $\{\mathbf{x}_j\}_{j \in \mathcal{N}_i}$, trains the SVM to synthesize the environment CBF with $\mathcal{Z}_i$ and $\{\mathcal{Z}_j\}_{j \in \mathcal{N}_i}$, and generates the action $\mathbf{u}_i^*$ with the CBF-CLF-QP controller. We can then consider $\{\mathbf{u}_i^*\}_{i=1}^n$ as the expert actions of the agents given the states $\{\mathbf{x}_i\}_{i=1}^n$ and available communication links. We thus consider the agent states $\mathbf{X}$ as features and the expert actions $\mathbf{U}^*$ as labels to build a training dataset in the framework of imitation learning. As we are interested in learning attentions $\mathbf{W}$ of graph edges, we consider a GAT $\mathbf{\Phi}_{\text{GAT}}(\mathbf{X}, \mathbf{W})$ to process features and formulate Problem 2 mathematically as an empirical risk minimization problem

$$\min_{\mathbf{W}} \ \frac{1}{M} \sum_{m=1}^{M} \ell(\mathbf{\Phi}_{\text{GAT}}(\mathbf{X}_m, \mathbf{W}), \mathbf{U}_m^*), \tag{8}$$

where $M$ is the number of training samples, $\ell(\cdot, \cdot)$ is the loss function that measures the difference between the predicted action and the expert action, and the edge weight matrix $\mathbf{W}$ is learned based on the attention mechanism. The learned $\mathbf{W}^*$ identifies the importance of the neighbors' states for the action generation of the local agent. I.e., a larger value of $[\mathbf{W}]_{ij}$ indicates that more information of the states $\mathbf{x}_j$ is used to generate the action $\mathbf{u}_i$ and thus, $A_j$ is more important to $A_i$.

**Thresholding.** While the trained GAT mimics an expert action, we need only the computed edge weights $\mathbf{W}^*$ to identify the neighbors' importance and leverage the latter to reduce the computation of SVM training. Consequently, we define a threshold $\gamma$ that determines if the LiDAR readings of a neighboring agent $A_j$ are used by the local agent $A_i$. When $[\mathbf{W}]_{ij} \geq \gamma$, the LiDAR readings of $A_j$ are important and will be used to train the SVM; when $[\mathbf{W}]_{ij} < \gamma$, the LiDAR readings of $A_j$ may be redundant, and will hence not be used in the computation.

## 5 Safety Guarantees for SVM-based CBFs

The proposed OE-CLBF estimates the unknown environment by training an SVM to formulate the corresponding CBF for collision avoidance. Different from conventional learning-based approaches, it is capable of providing safety guarantees with the accuracy bound inherited from SVM and the state invariance inherited from CBF. Specifically, let $\mathcal{Z} = \{(\mathbf{z}_l, y_l)\}_{l=1}^L$ be the dataset of $L$ LiDAR readings, with $\mathbf{z}_l \in \mathbb{Z} \subseteq \mathbb{R}^2$ and $y_l \in \mathbb{Y} = \{+1, -1\}$. For the space $\mathbb{Y}^{\mathbb{Z}}$, assume there exists an SVM classifier $f_{\text{SVM}}^*$ that classifies all samples in $\mathcal{Z}$ correctly. Define the error margin of $f_{\text{SVM}}^*$ as

$$\epsilon(f_{\text{SVM}}^*) = \min_{(\mathbf{z}_l, y_l) \in \mathcal{Z}} \frac{y_l <\boldsymbol{\omega}, \phi(\mathbf{z}_l) >_{\mathcal{F}}}{\|\boldsymbol{\omega}\|_{\mathcal{F}}}, \tag{9}$$

which measures the minimum distance of data samples to the classification hyperplane and identifies how well $f_{\text{SVM}}^*$ is w.r.t. $\mathcal{Z}$. For our safety analysis, we need the following assumptions.

**Assumption 1.** *The function $\phi$ of the SVM classifier is infinitely differentiable.*

**Assumption 2.** *There exists a constant $C$ s.t. for any $\mathbf{z} \in \mathbb{Z}$, it holds that $\Pr(\|\phi(\mathbf{z})\| \leq C) = 1$.*

Assumptions 1-2 are not restrictive as they are common for SVM classifiers [35, 33]. We now provide safety guarantees for the proposed method, and summarize the proof in Appendix D.

**Theorem 1.** *Consider the SVM-based CBF-CLF-QP controller with the function $\phi$ satisfying Assumptions 1-2 w.r.t. $C$. Let $\mathcal{C}_{\text{safe}}$ be the safety set and $\Delta t$ be the time interval between two successive LiDAR scanning, i.e., agents re-scan LiDAR readings every $\Delta t$. For any agent state $\mathbf{x}$ randomly distributed in $\mathcal{C}_{\text{safe}}$ with probability at least $1 - \alpha$, if the trained SVM $f_{\text{SVM}}^*$ classifies $L$ LiDAR readings, the agent state within the next time interval $\Delta t$ stays in $\mathcal{C}_{\text{safe}}$ at a large probability $1 - \delta$, where*

$$\delta \leq \frac{1}{L} \Big( \kappa(f_{\text{SVM}}^*) \log_2 \Big( \frac{8eL}{\kappa(f_{\text{SVM}}^*)} \Big) \log_2(32L) + \ln \Big( \frac{L^2}{\alpha} \Big) \Big) \text{ with } \kappa(f_{\text{SVM}}^*) = \Big\lceil \Big( \frac{8C}{\epsilon(f_{\text{SVM}}^*)} \Big)^2 \Big\rceil. \tag{10}$$

Theorem 1 states that the unsafety risk *is bounded by factors that are inversely proportional to the number of LiDAR readings $L$ and the square of error margin $\epsilon(f_{\text{SVM}}^*)^2$. In particular, $L$ depends*

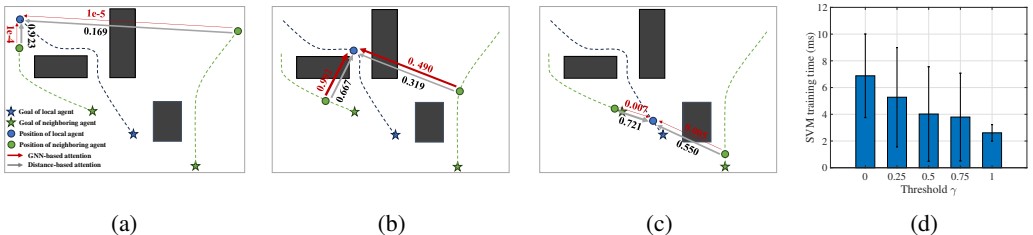

| (a) | (b) | (c) | (d) |

Figure 2. GAT-based attention and distance-based attention when: (a) Agents are close or far away; (b) Local agent needs information from its neighboring agents; (c) Agents are close to destinations. (d) Computation time of single SVM training with different thresholds $\gamma$ for attention-based computation reduction.

on the LiDAR scanner of the agent and $\epsilon(f^*_{\text{SVM}})$ is determined by the performance of the SVM classifier. When $L$ and $\epsilon(f^*_{\text{SVM}})$ becomes large, i.e., LiDAR scanner captures more readings from the environment and SVM classifier identifies two classes more clearly, the probability bound reduces towards zero and the unsafety risk decreases for multi-agent navigation. It is noteworthy that *this is the first learning-based CBF method with provable safety guarantees in a multi-agent setting*.[1]

## 6 Experiments

First, we provide a proof-of-concept for the proposed OE-CLBF, to show the effectiveness of SVM-based CBF-CLF-QP controller and attention-based computation reduction, in an environment with three agents and three obstacles. Then, we provide performance comparisons between our method and state-of-the-art baselines in environments with more agents and different obstacle layouts (i.e., more obstacles with different positions, sizes and shapes). For all experiments, we consider single integrator dynamics, and a Radial Basis Function (RBF) kernel for the SVM [38]. The method can be extended to general systems with complex dynamics, by substituting the SVM-based safety function (7) into corresponding CBF constraints [39, 40, 41], which will not increase the computation or required samples of the online SVM training. Additional experiments are provided in Appendix E.

**Proof of concept.** We consider an environment of size 8m×6m. The initial and goal positions **S** and **G** of the agents are distributed randomly in the environment. The agent has a circular shape of radius 0.1m with a maximum velocity 0.5m/s, and the obstacles are rectangles with different sizes. The sensing range of LiDAR scanner is 1.5m and the number of scanning lasers is 50. The maximal time step is 1000 and the time interval is 0.1s. For convenience of corroborating the relationship between the edge weights computed by GAT and the importance of neighboring agents, we assume a sufficiently large communication radius here with a fully connected graph among agents. We consider a distance-based attention mechanism as a baseline for comparison, which computes the edge weight that is inversely proportional to the distance between two neighboring agents.

Figs. 2a-2c plot agents' trajectories computed by our method with GAT-based attention at a threshold $\gamma = 0.5$. Our method completes all navigation tasks, and the removed LiDAR readings by attention-based thresholding do not affect CBF synthesis and navigation performance. There are three scenarios where GAT-based attention is small and computation can be saved: (*i*) Two agents are far away such that their LiDAR readings are irrelevant with useless information – see Fig. 2a; (*ii*) Two agents are close such that their LiDAR readings are similar and redundant – see Fig. 2a; (*iii*) The local agent is close to destination such that its navigation task is trivial and dose not need information from neighbors – see Fig. 2c. The distance-based attention is only small in scenario (*i*) and thus, does not capture all relevant cases. Moreover, Fig. 2b shows that GAT-based attention is large when LiDAR readings of neighbors are complementary and useful for the local agent. Fig. 2d shows the computation time of SVM training with different thresholds $\gamma$. For $\gamma = 0$, it trains the SVM with LiDAR readings of all neighbors and requires the most computation, while for $\gamma = 1$, it trains the SVM with only local LiDAR readings and requires the least computation. An appropriate selection of $\gamma$ (e.g., 0.5) saves almost half of the computation and maintains a good performance.

---

[1]The probability bound can be further reduced by leveraging improved accuracy analysis of SVM [36, 37].

Table 1: Performance comparison between our method and baselines in three metrics. A higher success rate, a higher SPL and a smaller number of time steps represent a better performance.

| Method \ Metric | Success rate | SPL | Time steps |
|---|---|---|---|
| **OE-CLBF (ours)** in unknown environments | **0.99 ± 0.02** | **0.83 ± 0.07** | **207.51 ± 36.76** |
| **Neural CLBF** in unknown environments | 0.19 ± 0.06 | 0.16 ± 0.08 | 760.94 ± 269.36 |
| **CBF-CLF-QP with RRT** in known environments | 0.98 ± 0.04 | 0.67 ± 0.07 | 458.50 ± 65.89 |
| **Global CBF-CLF-QP** in known environments | 0.85 ± 0.13 | 0.77 ±0.11 | 266.31 ± 111.39 |

**Performance evaluation.** We evaluate our method for a large system with 8 agents in an environment with 6 obstacles of different positions, sizes and shapes – see Fig. 1b for an example. The initial and goal positions are random, and the communication radius is 2.5m. We consider three state-of-the-art baselines: (*i*) Neural CLBF that leverages offline supervised learning to synthesize the CBF-CLF-QP controller and works in unknown environments [17, 19]; (*ii*) CBF-CLF-QP with RRT that requires environment knowledge, leverages it to generate collision-free trajectories with RRT, and follows these trajectories with CBF-CLF-QP [42, 43]; (*iii*) Global CBF-CLF-QP that requires environment knowledge to formulate CBFs for safe navigation [10, 44]. As it is non-trivial to formulate analytical CBFs for irregular obstacles even with environment knowledge, we approximate with circular obstacles of diameter as diagonal length. We use three metrics: success rate, Success weighted by Path Length (SPL), and time steps. The first is the ratio of successful agents to total agents, where success represents goal reaching without collision, the second is a stringent measure combining success rate and path length, and the third characterizes how fast agents reach destinations. We average results over 10 navigation tasks and 5 environments with 3 trials per case.

Table 1 shows the results. The proposed OE-CLBF consistently outperforms the baselines in all metrics, without requiring any knowledge of the environment. Neural CLBF performs worst although it also does not need the environment knowledge. This can be explained by the fact that offline supervised learning suffers from performance degradation when testing environments are significantly different from training ones, highlighting the importance of online SVM training. CBF-CLF-QP with RRT and global CBF-CLF-QP require full environment knowledge, which may not be practical and presented as benchmark values only for reference. CBF-CLF-QP with RRT achieves a comparable success rate to ours, but a lower SPL and larger time steps. This is because it randomly explores the entire space and may generate inefficient trajectories. Global CBF-CLF-QP has a lower success rate, but a larger SPL and smaller time steps compared to the one with RRT. We attribute this to the fact that it approximates rectangular obstacles with large circles and generates conservative agent behaviors, resulting in infeasible navigation scenarios for the CBF-CLF-QP controller [22].

## 7 Discussion

**Conclusion.** This paper focused on decentralized multi-agent navigation in unknown environments, and proposed the OE-CLBF controller consisting of two main components: (*i*) an SVM-based CBF-CLF-QP controller that estimates the environment by learning its corresponding CBF with an online SVM, based on instantaneous LiDAR readings, and generates safe actions by solving the resulting CBF-CLF QP; (*ii*) an attention-based computation reduction that predicts the importance of neighboring agents by learning edge weights with GATs, which allows the SVM to be trained with only relevant LiDAR readings. Our method provides provable safety guarantees, by leveraging an accuracy analysis of the SVM and state invariance of CBFs. We experimentally show that our approach achieves superior performance over baselines and adapts to a variety of unknown environments.

**Limitations.** While the attention-based computation reduction follows the intuition behind the inner-working mechanism of GATs, it is established upon the assumption that larger edge weights learned by the GAT represent higher importance of the corresponding neighbors, which has not been formally proved in theory. Additionally, the exploration mechanism incorporated in our method encourages agents out of local minima but still does not guarantee global success. Finally, we aim to extend our work with more complex non-linear dynamics such as unicycle models and drone modes.

**Acknowledgments**

This work was supported by European Research Council (ERC) Project 949940 (gAIa).

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

# A  Related Works

We provide more details about related works in the following comparison table, which categorizes prior approaches with respect to known v.s. unknown environments, known v.s. unknown dynamics and online v.s. offline learning. In our work, "environment" refers to the external configuration of the map that is independent of the agents, in particular it refers to the obstacle layout in the space (e.g., the obstacle position, shape and size). We consider agent dynamics and the communication capacity as internal properties of the agents, which are assumed known for our controller. This is a standard setting in the study of multi-agent navigation, particularly when dealing with unknown environments.

Table 2: Comparison table of related works

| Related Works | Known or Unknown Environment | Known or Unknown Dynamics | Online or Offline Learning |
|---|---|---|---|
| [10, 11, 12, 13, 14, 20, 21, 45, 46, 43, 47, 48, 49, 50] | Known | Known | \ |
| [25, 26, 27, 51] | Known | Known | Offline |
| [18, 17, 19, 23, 24] | Unknown | Known | Offline |
| [22] | Known | Known | Online |
| [52] | Unknown | Known | Online |

The majority of prior approaches focus on the setting with known environments (maps) and pre-defined dynamics to design safety-critical controllers for navigation. Learning-based approaches offer a way to leverage sensing inputs, including LiDAR readings and images, for safe navigation in unknown environments, while existing works mainly focus on offline supervised learning to synthesize controllers or estimate agent states. This suffers from performance degradation when testing scenarios are significantly different from training ones, and cannot provide safety guarantees due to the nature of end-to-end learning. The work in [22] leverages online learning to tune hyperparameters of the control barrier functions (CBF) in real-time but requires the knowledge of the map for deployment, while [52] studies a single-agent scenario in discrete grid maps and assumes the existence of a controller capable of following the discrete trajectory.

Note that all these works require the knowledge of system dynamics to either formulate analytic safety constraints or generate ground-truth labels for training. With this contextualization in mind, our work develops a sensing-feedback controller for multi-agent navigation in unknown environments. Our method learns the CBF of the surrounding environment with a support vector machine (SVM) in an online manner, enabling deployment in any new environment without prior knowledge. Moreover, we leverage graph attention learning to select important neighbors' states to reduce online computation and provide provable safety guarantees.

# B  Exploration Mechanism

While control inputs satisfying the CBF constraint always remain safe and control inputs satisfying the CLF constraint lead the agent to its goal, the QP in (5) becomes infeasible if these sets have an empty intersection. The latter causes agents to get stuck in complex navigation scenarios, such as environments with non-convex obstacle regions or where the destination is located directly on the opposite side of an obstacle [53, 19, 22]. To overcome this challenge, we propose an exploration mechanism inspired by the Expansive Search Tree (EST) [54] and integrate it into our method to encourage agents out of local minima. and integrate it into our method to ensure successful agent navigation.

Specifically, we consider the standard QP controller (5) as the convergence mode, and switch to the exploration mode when the QP controller becomes infeasible, i.e., no feasible control satisfies both convergence and safety constraints in (6). The exploration mode samples a state $\mathbf{x}_{\text{sample}} \sim \mathcal{P}_{\text{state},\text{d}_{\max}}$ from its nearby space with the sampling radius determined by the maximum scanning

---

**Algorithm 1** Exploration Mechanism (for agent $A_i$)

---

1: **Input:** State $\mathbf{x}_i$, target $\mathbf{x}_i^d$, safety constraints $\mathcal{S}$
2: **Output:** Control action $\mathbf{u}_i$
3: **if** QP controller infeasible **then**
4:       Switch to the exploration mode
5:       **for** $i_{\mathrm{explore}} = 1, \ldots, \mathrm{N}_{\mathrm{explore}}$ **do**
6:           **while** exploration mode **do**
7:               Sample a state $\mathbf{x}_{\mathrm{sample}} \sim \mathcal{P}_{\mathrm{state}, \mathrm{d}_{\max}}$
8:               Attempt to generate control action $\mathbf{u}_i$ to steer towards $\mathbf{x}_{\mathrm{sample}}$
9:               **if** $\mathbf{u}_i$ violates safety constraints $\mathcal{S}$ **then**
10:                  Discard $\mathbf{u}_i$
11:              **else**
12:                  break
13:          Apply $\mathbf{u}_i$ to system (1)
14:          Scan the environment to collect LiDAR readings
15:          Train SVM with the latest LiDAR readings to update the environment CBF
16:  Switch to the convergence mode

---

distance $d_{\max}$ of the LiDAR scanner, and generates an action by the QP controller to steer the agent towards the sampled state $\mathbf{x}_{\mathrm{sample}}$.

As we collect LiDAR readings at each time step and perform online SVM training, each agent estimates the surrounding environment in a real-time manner. For any generated action that violates safety constraints, the previously sampled state will be discarded and a new state will be sampled. After repeating this sampling process for a fixed number of times $\mathrm{N}_{\mathrm{explore}}$, we switch back to the convergence mode and navigates the agents towards their destinations. Algorithm 1 summarizes the exploration mechanism integrated in our method.

## C   Methodology Details

We provide additional details about the proposed method in this section.

### C.1   Graph Attention Networks

Graph attention networks (GATs) [32] are a variant of graph neural networks (GNNs) [55, 56, 57] that leverage the attention mechanism to extract task-relevant features from graph data. Different from conventional GNNs, which aggregate neighborhood information with (pre-defined) fixed weights, GATs allow to differentiate neighbors w.r.t. their contributions when aggregating neighborhood information to generate task-relevant features.

Specifically, consider a graph $\mathcal{G} = \{\mathcal{V}, \mathcal{E}\}$ with the node set $\mathcal{V} = \{1, \ldots, n\}$ and the edge set $\mathcal{E}$. The graph structure can be captured by a support matrix $\mathbf{W}$ with $(i, j)$th entry $w_{ij} = [\mathbf{W}]_{ij} \neq 0$ if node $i$ is connected to node $j$, i.e., $(i, j) \in \mathcal{E}$, or $i = j$, and $[\mathbf{W}]_{ij} = 0$ otherwise. The graph data can be captured by a graph signal $\mathbf{X}$, which is a matrix whose $i$th row $\mathbf{x}_i = [\mathbf{X}]_i$ contains the feature of node $i$. GATs are capable of assigning different attention coefficients $\{[\mathbf{W}]_{ij}\}_{j \in \mathcal{N}_i}$ to the neighboring nodes $\{j\}_{j \in \mathcal{N}_i}$ of each node $i$, which represent the importance of the neighbors' features for the feature update of the local node. These attention coefficients $\mathbf{W}$ are learned with a shared self-attention mechanism, enabling to quantify the relevance among graph nodes. Therefore, GATs are *suitable candidates* to characterize the importance of the neighbors' LiDAR readings in synthesizing the CBF of the unknown environment at the local agent.

The key component of the GAT lies in the attention mechanism, which computes attention coefficients between neighboring nodes. For the local node $i$ and its neighboring node $j$, the unnormalized attention coefficient is calculated as

$$e_{ij} = a(\mathbf{H}\mathbf{x}_i, \mathbf{H}\mathbf{x}_j), \tag{11}$$

where $\mathbf{H}$ is a learnable weight matrix (i.e., linear transformation) that maps the node features $\mathbf{x}_i$ and $\mathbf{x}_j$ to some higher-order features, and $a(\cdot)$ is a shared attention function such as a feedforward neural network. The unnormalized $e_{ij}$ is then passed through a softmax function to get the normalized attention coefficients as

$$w_{ij} = \frac{\exp(e_{ij})}{\sum_{k \in \mathcal{N}_i} \exp(e_{ik})}, \qquad (12)$$

which makes the sum of neighbors' attention coefficients equal to one for ease of probabilistic interpretation. With the normalized attention coefficients $\mathbf{W}$, each node $i$ generates its task-relevant feature as a weighted sum of neighbors' higher-order features

$$\mathbf{x}'_i = \sigma \Big( \sum_{j \in \mathcal{N}_i} w_{ij} \mathbf{H} \mathbf{x}_j \Big), \qquad (13)$$

where $\sigma(\cdot)$ is the non-linearity function such as ReLU and absolute value. Fig. 3 illustrates the attention mechanism, where the task-relevant feature, in our case, is the predicted action $\mathbf{x}'_i = \mathbf{u}_i$ that mimics the expert action $\mathbf{u}^*_i$ given in the training data.

**Discussion.** The attention coefficients learned by the GAT weigh the neighboring features when generating the task-relevant feature, i.e., the predicted action. That is, they represent how much the neighboring features are accounted for in the generation of the task-relevant feature. Therefore, it provides justifications for considering the attention coefficients as indicators to quantify the importance of the neighboring agents to the local agent, and for leveraging the latter to remove LiDAR readings of less important neighbors to reduce computation of online SVM training – see Section 4.2 of the main paper.

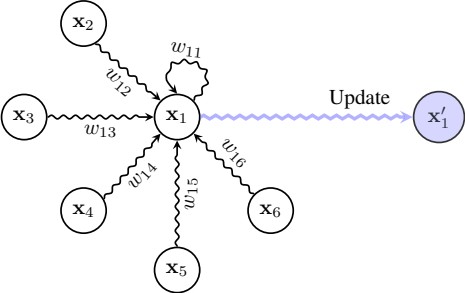

Figure 3. An illustration of the attention mechanism within the GAT [32]. The local node 1 aggregates the neighbors' features weighted by their corresponding attention coefficients, and leverages the latter to update its task-relevant feature $\mathbf{x}'_1$.

## C.2 Methodology Framework

The proposed method contains two main components: SVM-based CBF-CLF-QP navigation and attention-based computation reduction. The former generates safe actions for the agents to approach their destinations (Section 4.1 of the main paper), while the latter identifies important neighbors with the attention mechanism of the GAT and uses only LiDAR readings of these neighbors for online SVM training to reduce real-time computation (Section 4.2 of the main paper). Fig. 4 provides an overview for the general framework of the proposed OE-CLBF controller to ease understanding.

# D  Proof of Theorem 1

We start the proof by defining the generalization error of the learned SVM classifier $f^*_{\text{SVM}}$ as the probability risk in classifying any unseen data $(\mathbf{z}, y)$, i.e.,

$$R(f^*_{\text{SVM}}) = \Pr(f^*_{\text{SVM}}(\mathbf{z}) \neq y), \qquad (14)$$

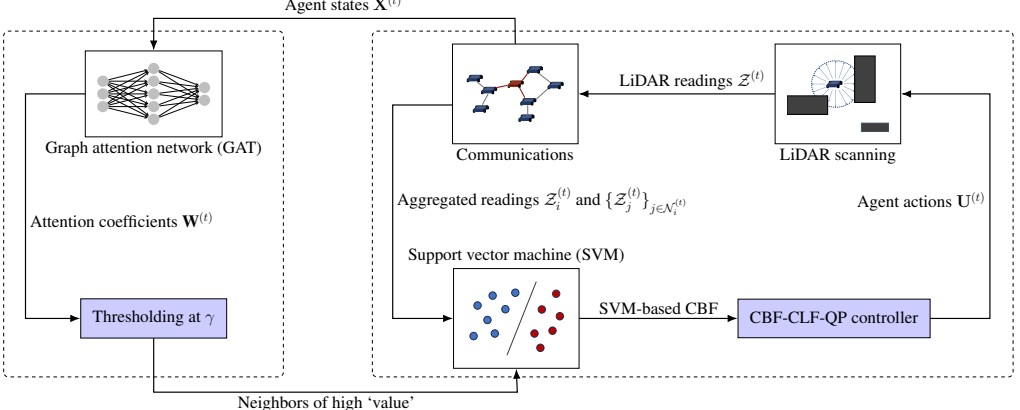

**Attention-based Computation Reduction**     **SVM-based CBF-CLF-QP Navigation**

Figure 4. General framework of the proposed OE-CLBF controller, which contains SVM-based CBF-CLF-QP navigation and attention-based computation reduction. The former senses the surrounding unknown environment with LiDAR scanner, synthesizes the corresponding CBF with online SVM, and incorporates it into the CBF-CLF-QP controller to generate agent actions. The latter quantifies the importance of the neighboring agents to the local agent with GAT, thresholds useless LiDAR readings from less important neighbors, and uses only useful and relevant LiDAR readings for online SVM trianing to reduce computation.

where $\Pr(\cdot)$ is the probability measure and $(\mathbf{z}, y)$ is drawn randomly from the space of interest. With the conditions provided in the theorem and the LiDAR readings $\mathcal{Z} = \{(\mathbf{z}_l, y_l)\}_{l=1}^L$, we can follow Theorem 1 of [33] to bound the probability risk as

$$R(f_{\text{SVM}}^*) \leq \frac{1}{L}\Big(\kappa(f_{\text{SVM}}^*)\log_2\Big(\frac{8eL}{\kappa(f_{\text{SVM}}^*)}\Big)\log_2(32L) + \ln\Big(\frac{L^2}{\alpha}\Big)\Big) \text{ with } \kappa(f_{\text{SVM}}^*)) = \Big[\Big(\frac{8C}{\epsilon(f_{\text{SVM}}^*)}\Big)^2\Big]. \tag{15}$$

For the learned safety function $h_{\text{SVM}}^*$ from $f_{\text{SVM}}^*$, (15) is equivalent to that for any position $\mathbf{z} \notin \mathcal{Z}$, if $h_{\text{SVM}}^*(\mathbf{z}) > 0$, it holds that

$$\Pr(\mathbf{z} \in \mathcal{C}_{\text{safe}}) \geq 1 - \delta \tag{16}$$

with

$$\delta \leq \frac{1}{L}\Big(\kappa(f_{\text{SVM}}^*)\log_2\Big(\frac{8eL}{\kappa(f_{\text{SVM}}^*)}\Big)\log_2(32L) + \ln\Big(\frac{L^2}{\alpha}\Big)\Big) \text{ with } \kappa(f_{\text{SVM}}^*)) = \Big[\Big(\frac{8C}{\epsilon(f_{\text{SVM}}^*)}\Big)^2\Big], \tag{17}$$

where $\mathcal{C}_{\text{safe}}$ is the safety set of the environment. Define the safety set $\mathcal{C}_{\text{SVM,safe}}$ w.r.t. the learned safety function $h_{\text{SVM}}^*$ as

$$\mathcal{C}_{\text{SVM,safe}} = \{\mathbf{z} | h_{\text{SVM}}^*(\mathbf{z}) > 0 \text{ for all } \mathbf{z} \in \mathcal{E}\}. \tag{18}$$

By substituting this safety function $h_{\text{SVM}}^*$ into the CBF constraint (3) and the latter into the CBF-CLF-QP controller (6), we can follow Theorem 3 of [12] to prove that there exists a class $\mathcal{K}$ function $\alpha(h_{\text{SVM}}^*)$ such that the $h_{\text{SVM}}^*$-based safety set $\mathcal{C}_{\text{SVM,safe}}$ [cf. (18)] is *forward invariant*, i.e., the agent will stay in $\mathcal{C}_{\text{SVM,safe}}$ with generated actions.

Since the $h_{\text{SVM}}^*$-based safety set $\mathcal{C}_{\text{SVM,safe}}$ depends on the learned safety function $h_{\text{SVM}}^*$, which, in turn, depends on the training data of $L$ LiDAR readings $\mathcal{Z}$, and each agent only re-scans these LiDAR readings every $\Delta t$, we assume $\mathcal{C}_{\text{SVM,safe}}$ keeps the same in the time interval $\Delta t$. Therefore, we have

$$\mathbf{z}^{(t)} \in \mathcal{C}_{\text{SVM,safe}} \tag{19}$$

for any time $t$ until the next time interval $\Delta t$.

By combining (16) and (19), we complete the proof that

$$\Pr(\mathbf{z}^{(t)} \in \mathcal{C}_{\text{safe}}) \geq 1 - \delta \tag{20}$$

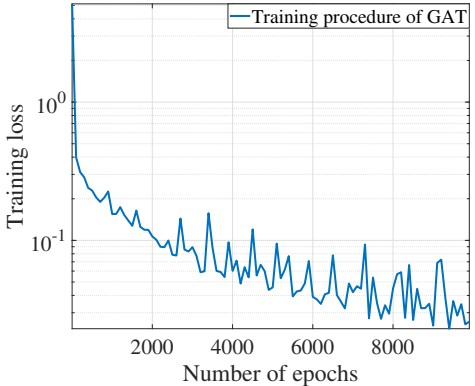

Figure 5. The training procedure of graph attention network (GAT) in the framework of imitation learning.

with

$$\delta \leq \frac{1}{L}\Big(\kappa(f^*_{\mathrm{SVM}})\log_2\Big(\frac{8eL}{\kappa(f^*_{\mathrm{SVM}})}\Big)\log_2(32L)+\ln\Big(\frac{L^2}{\alpha}\Big)\Big) \text{ with } \kappa(f^*_{\mathrm{SVM}}))=\Big\lceil\Big(\frac{8C}{\epsilon(f^*_{\mathrm{SVM}})}\Big)^2\Big\rceil \quad (21)$$

for any time $t$ in the next interval $\Delta t$.

## E  Additional Experiments

We conduct additional experiments to further validate methodology and corroborate theory.

### E.1  SVM-based Safety Function

The goal of this experiment is to verify the accuracy of the SVM classifier for learning the safety region of the unknown environment. Specifically, we train the SVM classifier with different numbers of random LiDAR readings from the environment in Fig. 1b, and evaluate the decision function of the trained SVM on 100 randomly sampled (unseen) LiDAR readings over the entire environment.

Table 3 shows the results. We see that the evaluation accuracy increases and approaches one as the number of training samples increases. This indicates that the generalization error of the SVM-based safety function approaches zero with the number of training samples, which corroborates our theoretical result in Theorem 1. Moreover, the SVM-based safety function achieves good performance even with a small number of training samples (e.g., 250), which highlights the strong classification capacity of the SVM.

Table 3: Accuracy of the SVM-based safety function

| Number of Training Samples | 250 | 500 | 1000 | 2500 | 5000 |
|---|---|---|---|---|---|
| Testing Accuracy | 0.87±0.04 | 0.89±0.02 | 0.90±0.04 | 0.93±0.05 | 0.97±0.02 |

### E.2  Graph Attention Learning.

The goal of the following experiments is to further validate the effectiveness of the attention mechanism in Section 4.2. *First*, we show the training procedure of the graph attention network (GAT), which mimics the expert action, i.e., the one generated with all neighborhood information, following an imitation learning framework. The initial and goal positions $\mathbf{S}$ and $\mathbf{D}$ of the agents are initialized randomly in the environment and the obstacles are distributed between $\mathbf{S}$ and $\mathbf{D}$. We generate 200 multi-agent trajectories of 200 time steps with $4 \times 10^4$ samples to construct the dataset, and leverage the ADAM optimizer with a learning rate $10^{-4}$ for training. The mean square error (MSE) is used as the loss function to measure the difference between the predicted action and the expert action.

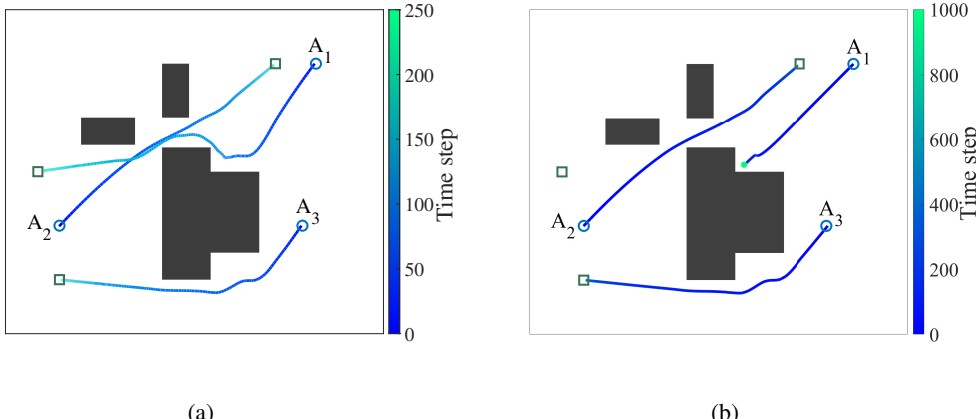

(a)                                      (b)

Figure 6. (a) Navigation procedure of our method with exploration mechanism, where all agents complete their tasks successfully. (b) Navigation procedure of our method without exploration mechanism, where agent $A_1$ gets stuck in local minima because of limited sensing and communication ranges. Blue circles are initial positions and green squares are destinations. Black rectangles are obstacles and blue-to-green lines are agent trajectories. Color bar represents time scale, showing that no agent-agent nor agent-obstacle collision occurs.

Fig. 5 plots the training procedure of the GAT over $10^4$ epochs. We see that the training loss decreases with the number of epochs and the decreasing rate reduces; ultimately, reaching a stationary solution. The convergent loss is small, which indicates a good performance of the trained GAT for action generation and a good prediction of attention coefficients for neighbors' importance.

*Second*, we emphasize the performance of the GAT by comparing with a heuristic method inspired by the attention learning mechanism in [58][2], in addition to the distance-based baseline considered in Section 6. Specifically, we follow the idea of spatial confidence-aware communication in [58] to develop a heuristic method to act as a strong baseline, i.e., we assuming that agents have less confidence in areas where there is no LiDAR detection and prompt communication in such regions.

Table 4 provides the comparison results. The proposed method achieves the best performance with the least computation time of the online SVM training. The spatial confidence-based baseline performs worse than the proposed method in all three metrics because our GNN-based attention better captures the importance of neighbors' states. Moreover, the online SVM training time of the spatial confidence-based baseline is higher than our method because: (i) some less-confident areas may be irrelevant to agents' navigation and it is not necessary to provide estimation for these areas; (ii) some agents in the direction of less-confident areas are too far from the local agent and their LiDAR readings are irrelevant to the local navigation. Both facts result in unnecessary LiDAR readings and more online computation. The distance-based baseline obtains the worst performance as it computes attention only based on the distance between two agents without considering their LiDAR reading relevance. Lastly, we remark that any attention computing mechanism can be applied to our SVM-based CBF-CLF-QP framework, to develop extended variants for further performance improvement.

### E.3 Exploration Mechanism.

The goal of this experiment is to show the necessity of incorporating the exploration mechanism [Appendix B] into our method for resolving challenging navigation scenarios. Fig. 6 shows one example of these scenarios, where the obstacles form a non-convex safety space and the agent easily

---

[2]The attention learning mechanism in [58] cannot be applied directly to our SVM-based CBF-CLF-QP controller. Specifically, it computes attentions among agents via end-to-end learning for a specific downstream task, where attentions are neural network outputs trained using a well-defined task-specific loss function (e.g. the detection loss based on given ground-truth labels in a supervised manner). However, in our case, it is challenging to define the loss function for training because it is non-trivial to measure how well the generated attention values quantify the neighbors' importance w.r.t. the proposed SVM-based CBF-CLF-QP controller. Therefore, there is no end-to-end loss function we can use and [58] can not be applied directly in our setting.

Table 4: Performance of our attention mechanism and baselines

| Method | Success Rate | SPL | Time Step | SVM Training Time (ms) |
|---|---|---|---|---|
| OE-CLBF (Our Method) | 0.99±0.02 | 0.83±0.07 | 207.51±36.76 | 4.01±3.53 |
| Spatial Confidence-based Baseline | 0.81±0.17 | 0.71±0.18 | 396.71±180.87 | 4.48±3.24 |
| Distance-based Baseline | 0.78±0.20 | 0.70±0.19 | 380.51±198.27 | 4.54±3.04 |

gets stuck in a local region due to its limited sensing and communication ranges. We see that our method navigates all agents towards destinations successfully in Fig. 6a, where the exploration mechanism pulls agent $A_1$ out of the local dilemma by searching over its nearby space. The method without exploration mechanism fails the navigation task of agent $A_1$ in Fig. 6b, because it can only generate actions that stay in the local safety set.

## E.4   Ablation Study

The goal of the following experiments is to perform an ablation study for the proposed method. *First*, we evaluate our method with different cut-off thresholds for the attention mechanism and show the results in Table 5. The performance of the proposed method increases as the value of threshold decreases. This is because a lower threshold allows for more communication among agents, thereby providing each agent with more LiDAR readings from its neighbors, which in turn leads to more accurate CBF estimates with the SVM and improves the navigation performance.

Table 5: Performance with different cut-off thresholds for the attention mechanism

| Threshold \ Metric | Success Rate | SPL | Time Step |
|---|---|---|---|
| 0.3 | 0.99±0.02 | 0.85±0.11 | 199.34±73.18 |
| 0.4 | 0.99±0.02 | 0.83±0.12 | 208.03±84.31 |
| 0.5 | 0.99±0.02 | 0.83±0.07 | 207.51±36.76 |
| 0.6 | 0.97±0.05 | 0.81±0.09 | 210.95±65.69 |
| 0.7 | 0.94±0.08 | 0.78±0.13 | 230.83±102.74 |

*Second*, we evaluate our method without the exploration mechanism and present the results in Table 6. The SVM-based QP controller performs worse without the exploration mechanism in all three metrics. This can be explained by the infeasibility of the SVM-based CBF-CLF-QP controller in certain challenging navigation scenarios, as discussed in Appendix B.

Table 6: Performance with and without the exploration mechanism

| Metric | Success Rate | SPL | Time Step |
|---|---|---|---|
| SVM-based CBF-CLF-QP with exploration | 0.99±0.02 | 0.83±0.07 | 207.51±36.76 |
| SVM-based CBF-CLF-QP without exploration | 0.68±0.24 | 0.62±0.22 | 426.29±208.36 |

## E.5   Method Robustness

The goal of the following experiments is to study the robustness of the proposed method to real-world deployment effects, i.e., the update frequency of online LiDAR readings and the sensor noise on LiDAR readings.

*First*, we set the update time as $0.1$s in our work, which is a reasonable update frequency for practical systems with LiDAR scanners [59, 60]. Table 7 shows the performance of the proposed method with different update times of online LiDAR readings. We see that the performance degrades as the update time increases (i.e., the update frequency decreases) as expected. This is because a low frequency leads to a delay in LiDAR reading updates and the CBF constraint learned by the

SVM may be outdated, resulting in worse environment estimation and performance degradation. However, the proposed method maintains good performance for relatively low update frequencies, which validates the robustness of the proposed method to update frequencies.

Table 7: Performance with different update times of online LiDAR readings

| Update Time (s) \ Metric | Success Rate | SPL | Time Step |
|---|---|---|---|
| 0.1 | 0.99±0.02 | 0.83±0.07 | 207.51±36.76 |
| 0.15 | 0.95±0.08 | 0.79±0.10 | 217.55±62.73 |
| 0.2 | 0.92±0.10 | 0.77±0.14 | 263.06±107.02 |
| 0.25 | 0.89±0.12 | 0.74±0.14 | 289.06±108.85 |
| 0.3 | 0.84±0.10 | 0.72±0.13 | 310.85±91.73 |

*Second*, we evaluate the proposed method with different levels of random noise on LiDAR readings of the agents, where the imposed noise on each axis (i.e., x-axis and y-axis) is sampled randomly from the interval $[0, \beta]$ and $\beta$ is the parameter that controls the noise level with the unit as meter. Table 8 shows the results. We see that the performance of the proposed method degrades with the increase of noise level. This is expected because larger noise on LiDAR readings results in less accurate estimation of the SVM-based CBF constraint. However, the proposed method remains robust to moderate noise and maintains good performance in all cases. Lastly, we clarify that the magnitudes of the noise considered in these experiments are reasonable in practical systems [61, 62], and techniques have been developed to alleviate the noise on LiDAR readings [63, 64].

Table 8: Performance with different levels of sensor noise on LiDAR readings

| Noise Level $\beta$ (m) \ Metric | Success Rate | SPL | Time Step |
|---|---|---|---|
| 0 | 0.99±0.02 | 0.83±0.07 | 207.51±36.76 |
| 0.005 | 0.95±0.08 | 0.78±0.10 | 223.55±79.33 |
| 0.01 | 0.93±0.16 | 0.76±0.16 | 249.93±144.02 |
| 0.025 | 0.94±0.08 | 0.74±0.14 | 254.95±105.70 |
| 0.05 | 0.91±0.13 | 0.74±0.15 | 256.25±123.27 |
| 0.1 | 0.86±0.19 | 0.72±0.16 | 301.18±165.90 |

### E.6 More Examples.

Figs. 7a-7f show more examples of agents' trajectories generated by our method, for different navigation tasks in unknown environments with different obstacle positions, sizes and shapes. We see that in all these scenarios, all agents complete their navigation tasks successfully, where the success represents the agents reaching their destinations without collision within the maximal time step, and do not require any prior knowledge about environments, even in some challenging scenarios with cluttered obstacles. This highlights the effectiveness of the proposed OE-CLBF controller, which is applicable in unknown environments with different obstacle layouts.

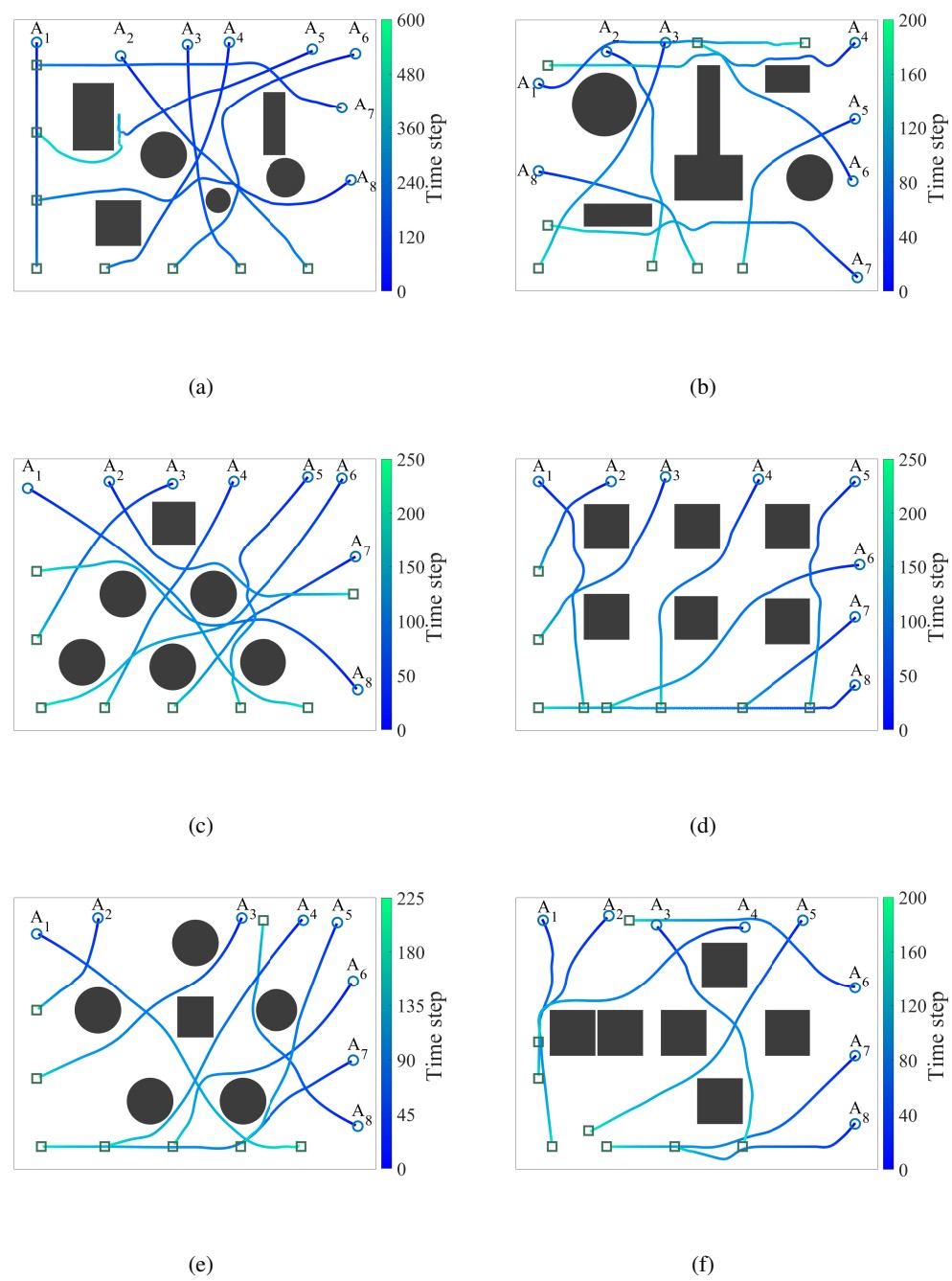

Figure 7. Navigation procedures of our method for different navigation tasks in different unknown environments with different obstacle layouts (i.e., different obstacle positions, sizes and shapes), and our method is capable of completing all navigation tasks successfully without collision either to obstacles or among agents.

