# OpenReview forum: "Provably Safe Online Multi-Agent Navigation in Unknown Environments"
_robot-learning.org/CoRL/2024/Conference — CoRL 2024_

### Official Review · Reviewer_tTRi · 2024-07-15
**Review of Provably safe online multi-agent navigation in unknown environments**

**Originality:** 4
**Technical Quality:** 5
**Clarity Of Presentation:** 5
**Potential Impact:** 3
**Recommendation:** 4
**Confidence:** 3

**Review:**

**Quality and Clarity**

The quality of the paper is very high and its ideas are clearly well written. I can not verify all theoretical contributions but the
math was sound as far as I followed it.

**Originality**

The work contributes theoretical and framework advances by incorporating SVMs and GATs.
The work seems to be original and thorough work of the authors.

**Significance**

The work has medium significance as it shows how to use GATs to reduce computational time, can be used as a safe component in larger framework or could be improved upon on the theoretical side. However, none of these seem to be large advances.

**Strengths**
Good Robotic Experiments are done with a multitude of provided scenarios. Testing against Baselines was done and fair comparisons to state of the art methods are established.
The theoretical advances seem to be well done and contribute to the scientific literature.
The quality of the writing is throughout the paper high and its main ideas are clearly stated and experiments covered all laid out claims in the introduction.

**Weaknesses**
None came to mind reading the paper

**Quality Of The Limitations Section:**

3

**Questions For Rebuttal:**

I have no questions to the authors regarding submission.

**Robotics Focus:**

4

**Summary Of Paper:**

The paper addresses the challenge of decentralized multi-agent navigation in unknown environments by proposing an Online Exploration-based Control Lyapunov Barrier Function controller. The OE-CLBF overcomes the limitation of full environment knowledge by learning Control Barrier Functions using online Support Vector Machine training with local LiDAR data. The paper contributes an SVM based method for learning CBFs in real time, an graph attention network to reduce computational load and it provides a safety guarantee.

**Summary Of Recommendation:**

The paper seems to be done with the necessary due dilligence done and was a joy to read.

---

### Official Review · Reviewer_nmUB · 2024-07-19
**More clarity needed**

**Originality:** 3
**Technical Quality:** 2
**Clarity Of Presentation:** 2
**Potential Impact:** 2
**Recommendation:** 2
**Confidence:** 4

**Review:**

While the work presents some interesting ideas, there are several issues with clarity, technical details, and practical applicability that need to be addressed.

Major concerns:

1. The related work discussion lacks sufficient depth and organization. A comparison table categorizing prior approaches (e.g. known vs unknown map, known vs unknown dynamics, online vs offline) would help contextualize this work.

2. The definition of "unknown environment" needs clarification. Only the map layout seems unknown, while agent dynamics are known and communication is allowed. This is a much weaker "unknown environment" setting than typically referred to.

3. The complexity of agent dynamics is not adequately addressed. More complex dynamics can significantly complicate CBF learning.

4. Many mathematical symbols are introduced without proper explanation (e.g. inf in Eq. 2, $\delta$ in Eq. 6). The notation needs to be more carefully organized.

5. The key insights and advantages of the approach over existing methods should be more clearly articulated, beyond just stating "provable guarantees". For example, Theorem 1 lacks clarity, with undefined terms ($\kappa$) and insufficient discussion of parameters like $C$, neither theoretically nor empirically. In practice, $C$ can be extremely large which makes the conclusion meaningless.


Practical concerns:

1. Compressing LiDAR data to binary values (-1, 1) discards valuable depth information. The impact of this simplification is not justified.

2. The real-time feasibility and update frequency of the algorithm are not discussed in experiment, which is crucial for collision avoidance. Low frequency may cause still cause collision because the agent cannot respond in real-time.

3. The graph attention component requires expert demonstrations, it is unclear how to get this data in advance.

4. How robust is your approach to sensor noise, particularly for LiDAR inputs? Given that real LiDAR sensors typically have some level of noise, and this is a major challenge when deploying to real robots, can you provide evidence or analysis demonstrating that your method can handle noisy input data? Additionally, since no real robot experiments were presented, what assurances can you offer that this approach is viable for real-world deployment?

It is recommended to provide a demo video to better illustrate the approach.

**Quality Of The Limitations Section:**

2

**Questions For Rebuttal:**

- How does this work compare to and improve upon existing approaches for multi-agent navigation in unknown environments, particularly in terms of the assumptions made about the environment, agent dynamics, and available information? A clearer comparison and positioning relative to prior work seems needed.

- What is the real-world feasibility and robustness of this approach, especially regarding computational requirements, update frequency, sensor noise handling, and applicability to more complex agent dynamics? More details on practical implementation concerns appear necessary.

- Can you provide more clarity and justification for some of the key technical elements, such as the binary encoding of LiDAR data, the theoretical guarantees in Theorem 1, and the process for obtaining expert demonstrations for the graph attention component?

**Robotics Focus:**

2

**Summary Of Paper:**

This paper proposes an Online Exploration-based Control Lyapunov Barrier Function (OE-CLBF) controller for safe decentralized multi-agent navigation in unknown environments.

**Summary Of Recommendation:**

While this work may have potential contributions to safe multi-agent planning and control, the current lack of clarity and addressing of practical concerns make it difficult to recommend for acceptance in its present form. Substantial revisions are needed to improve the presentation and address the raised issues.

---

### Official Review · Reviewer_wuRr · 2024-07-21
**Review of CoRL submission 306**

**Originality:** 3
**Technical Quality:** 3
**Clarity Of Presentation:** 2
**Potential Impact:** 3
**Recommendation:** 3
**Confidence:** 3

**Review:**

The motivation for multi-agent collision avoidance in an unknown environment is sound and clear. The paper is well-written and easy to follow. I like the idea of incorporating SVM into CBF-CLF-QP, which is insightful and SVM-based CBF can be updated in an online manner. The theoretical guarantee is sound. Here are some weaknesses.

- I wonder if the SVM-based CBF is differential when solving QP problem (6) as the gradient of Eq (8) wrt $x$ is needed for the Lie derivative. How to connect LiDAR readings $z$ with state $x$?

- The infeasibility of QP is resolved by exploration, which is an empirical solution. Is there any theoretical perspective regarding the infeasibility of SVM-based QP?

- Although Thm 1 is sound, it is unknown how tight the bounds are. It seems that the convergence rate wrt $L$ is $\mathcal{O}(\frac{\log L}{L})$ and the convergence rate wrt $\epsilon$ is $\mathcal{O}(\frac{\log \epsilon}{\epsilon^2})$, which are only asymptotic and not very tight. Is there any tighter bound to achieve exponential convergence? Also, since the probabilistic guarantee is not sound, can the tightness be verified through Monte Carlo sampling in the experiment?

- For the experiment, the distance-based attention is too naive and may not be convincing. It is expected to compare the proposed method with other baselines from the literature. Also, a quantitative comparison is expected besides qualitative comparison in Figure 2. In Figure 2, it seems that the proposed is unnormalized but the distance-based attention baseline is somehow normalized, so it needs more clarification to justify the results. Time comparison with the baseline should be reported in Figure 2d as well.

- More ablation study over the proposed method is needed, e.g. to show the effectiveness of attention, exploration mechanism in Alg 1.

- Type on line 164: there is no reference to Fig. 3a.

**Quality Of The Limitations Section:**

3

**Questions For Rebuttal:**

See weakness.

**Robotics Focus:**

3

**Summary Of Paper:**

This paper studies multi-agent safe navigation in unknown environment, and introduces an onlinear learning of CBF with SVM using LiDAR sensing. GAT is adopted to predict the importance of the neighbor agents. Experiments validate the effectiveness of the proposed method.

**Summary Of Recommendation:**

I would be happy to increase my rating if the theory details are clarified and more experiments are conducted as I mentioned in weakness.

---

### Author Rebuttal · Authors · 2024-08-12

We have uploaded three video demonstrations for different navigation scenarios with the proposed method to a zip file. Thank you!

---

### Decision · Program_Chairs · 2024-09-04

**Decision:**

Accept

**Comment:**

This paper studies multi-agent safe navigation in an unknown environment and introduces an online learning of CBF with SVM using LiDAR data. GAT is adopted to predict the importance of the neighbor agents.

Strengths
- The motivation for safe multi-agent navigation in an unknown environment is sound and clear.
- The idea of incorporating SVM into CBF-CLF-QP is interesting, especially considering its potential in updating the CBF in an online manner.

Limitations
- The method requires further comparison (both qualitative and quantitative) with existing approaches for multi-agent navigation in unknown environments, particularly in terms of the assumptions made about the environment, agent dynamics, and available information.
- The authors should properly discuss the feasibility and applicability of the proposed method in terms of more general dynamical systems, computational requirements, and sensor noise. Quantitative analysis in this regard would further strengthen the paper.
- More ablation studies and justification is required for some of the key technical elements, such as to highlight the utility of binary encoding of LiDAR data, to show the effectiveness of attention, exploration mechanism in Alg 1, etc.

**Comments post rebuttal**

The additional experiment results make the contribution of this work more clear and convincing. It would be good to include these results as well as a discussion on the online computational complexity of the proposed algorithm in the final version of the paper. There are some additional small suggestions by reviewers that could be incorporated to further improve the paper clarity and exposition.